# Relations between Positive Parenting Behavior during Play and Child Language Development at Early Ages

**DOI:** 10.3390/children10030505

**Published:** 2023-03-03

**Authors:** Magda Rivero, Rosa Vilaseca, María-José Cantero, Clara Valls-Vidal, David Leiva

**Affiliations:** 1Department of Cognition, Developmental and Educational Psychology, University of Barcelona, 08035 Barcelona, Spain; 2Department of Developmental and Educational Psychology, University of Valencia, 46010 Valencia, Spain; 3Department of Psychology, University Abat Oliba-CEU, 08022 Barcelona, Spain; 4Department of Social Psychology and Quantitative Psychology, University of Barcelona, 08035 Barcelona, Spain

**Keywords:** parenting, positive parenting, parental behaviors, fathers, adult–child interaction, child language development, responsiveness

## Abstract

Parental behavior in interactions with children has been related to child language development. Our study contributes to the literature about relations between the characteristics of parent–child interactions during play and a child’s language development in typically developing children at early ages, with data from mothers and fathers from the same families in Spain. Our aim was to analyze the relation between positive parenting behaviors assessed with the Spanish version of the Parenting Interactions with Children: Checklist of Observations Linked to Outcomes (PICCOLO) and child language development assessed with the Bayley-III scales. We controlled for some sociodemographic variables. The participants were 90 children aged 15–31 months and their mothers and fathers. Bivariate analysis showed significant positive relations between mothers’ responsive, encouraging and teaching behaviors and a child’s language scores. Relations were found between fathers’ encouraging and teaching behaviors and a child’s language. Regression models indicate that maternal and paternal encouraging behaviors predicted 18% of the variability in the child’s receptive language, and maternal responsive and teaching behaviors predicted 16% of the variability in the child’s expressive language and total language scores. The study provides new data that support the relevance of positive parental behaviors to improve a child’s linguistic development.

## 1. Introduction

One of the areas of greatest interest researched in relation to child development is which parenting characteristics are associated with variability in children’s development outcomes. From a sociocultural and systemic approach [1,2,3,4], child development is conceptualized as a process that is intrinsically linked to the characteristics of the contexts in which it occurs and to the interpersonal processes that take place in these contexts. More specifically, from these systemic and interactive models, it is proposed that child development is the product of interaction of children with their parents and other primary caregivers in daily routines [4,5,6]. From this point of view, it is claimed that variability in child development can be explained not only by children’s individual constitutional characteristics but also by quality of parenting. Quality and quantity of parental behaviors could make a difference in early childhood development [7,8].

With respect to linguistic development, it is well established in the literature that some variables that define socioeconomic status (SES), particularly maternal and paternal educational levels and household income, relate to a child’s communicative and linguistic outcomes [9,10,11,12]. Positive relations between these variables have been explained in terms of family investment and family stress [11]. With respect to family investment, high economic resources would permit greater investment in materials and experiences that promote children’s development [13,14] and a parent’s high educational level would be related to more responsive, stimulating parental behaviors, which may in turn promote better communicative and linguistic development of the child [15,16]. Family stress has been proposed as another factor explaining the relations between SES and children’s communicative and linguistic development. Family income may impact the parents’ psychological well-being, which in turn would affect parental behaviors in childrearing [14] and the child’s language development [11].

In addition to SES, at a more microanalytical level, which is the focus of our work, some qualitative and quantitative properties of early adult–child dyadic interactions, including adults’ speech addressed to the child, play a relevant role in communication and language development [17,18,19,20]. The study of language development is a key topic in disciplines that deal with child development and education. It would be interesting to identify factors linked to optimal linguistic development or that can be protective factors for linguistic delay to advance in the understanding of the process of language acquisition and to contribute to sustaining intervention evidence-based practices.

In an analysis of the properties of caregiver–child interactions that promote a child’s development, the caregiver’s sensitivity or responsivity (responsiveness) appears to be one of the most relevant features. Ainsworth et al. [21] defined maternal sensitivity as the mother’s ability to adequately perceive and interpret the signals and communications implicit in the child’s behavior and to provide appropriate, quick responses. They linked the caregiver’s sensitivity to a child’s secure attachment.

With respect to language, a review of the literature reveals that a caregiver’s responsivity (or responsiveness) during dyadic interactions is one of the most relevant, consistent factors linked to a child’s linguistic outcomes at early ages [22,23,24,25,26]. The most common definition of responsiveness coincides with that offered by Ainsworth et al. [21] on sensitivity. A responsive or sensitive caregiver is attentive to the child’s signals about his/her emotional states, motivations and interests, interprets those signals and responds to them in a contingent way. In other words, they give the child a fast, adjusted response [27]. Contingent responses to the child’s focus of attention and to his/her actions have been related to better child’s linguistic outcomes [22,28,29]. Tamis-LeMonda et al. [27] offered a theoretical explanation for the role of a caregiver’s responsiveness in child language development. First, the close relation between the child’s behavior, interests or emotions and the adult’s response (contingency) helps the child to establish relationships between language and the world. Additionally, the adult’s responses usually incorporate speech, whether in the form of comments, labels, questions, suggestions, commands or descriptions. Thus, the responses provide linguistic models. Finally, use of nonverbal behavior that frequently accompanies speech (such as pointing or looking) facilitates language comprehension and learning. In addition, from a neurobiological approach, a sensitive, responsive parent operates within the child’s zone of proximal development and thereby builds the neural architecture for joint attention and language [30].

Some authors have proposed a broader definition of caregiver’s sensitivity that includes other emotional components of adult–child interaction, such as warmth, affection, timing, flexibility, acceptation and negotiation [31] or the caregiver’s interest in promoting the child’s motivation and cooperation during play in a climate of positive affection [32]. In a recent review of the literature, Deans [33] pointed out that an improvement in child’s comfort, emotional support, emotional availability and emotional expressiveness are commonly considered relevant characteristics of sensitive childrearing. Many studies have evidenced that children are very sensitive to an adult’s expression of emotions and to the synchrony of the mother’s responses to their behavior from early ages [34]. Infants also prefer expression of positive emotions when they interact with an adult [35]. Accordingly, affect appears to be a crucial component of responsive behavior [22]. As Dave et al. [22] have pointed out, studies that link responsiveness and child language development typically consider affect and infant-directed speech in an overall measure of responsiveness. A recent metanalytic study that examines parental behavior and early childhood language in typically developing samples indicates that children whose caregivers show higher levels of sensitivity and warmth display stronger language skills than children who receive lower levels of such parenting behavior [25].

Another dimension of caregivers’ behavior that has been considered with respect to language development is directiveness. Parental directiveness is an intrusive, controlling interaction style characterized by a tendency to make commands or statements to the child that are unrelated to the child’s focus of attention or action [36]. Contrary to responsiveness, parental restrictive, punitive, intrusive and controlling behaviors [37,38,39] and directive speech [40] have been linked to poorer language development outcomes. Nevertheless, a difference between contingent directiveness and intrusive directedness must be made. Contingent directives are those related to the child’s focus of attention (e.g., a command about what to do with the object the child is attending to). They have not been linked to poorer linguistic development trajectories [41,42].

Finally, parental behavior that supports a child’s language development includes the way adults structure their conversational interchanges with children and some characteristics of child-directed speech. Quantitative and qualitative properties of child- or infant-directed speech are relevant to a child’s language acquisition [43]. Weisleder and Fernald [44] found that the quantity of speech addressed to a child benefits their vocabulary. Huttenlocher et al. [10] related the quantity and diversity of input to a child’s vocabulary and syntactic growth. Use of diverse, rich vocabulary in interactions with a child and the presence of explanations (logical connections between objects or events), language accompanying symbolic play and talk about past and future predict better vocabulary outcomes [43]. Partial or total repetitions of the child’s vocalizations or utterances also improves language acquisition at early ages [45,46]. Repetitions of the child’s utterance, adding syntactic or semantic information, usually named “expansions” or “extensions”, respectively, positively correlates with the child’s conversational skills and semantic and syntactic development [47]. Questions have also been related to a child’s vocabulary [48,49].

Research on parental behavior that is positively related to a child’s language developmental outcomes is generally very specific and focuses on a reduced group of behavior using systems of ad hoc categories for the analysis. This has been a good research strategy as it has allowed a large volume of knowledge about parental behavior to be compiled that contributes to optimizing child language development. Nevertheless, in professional contexts of intervention with families, microanalytical strategies for interaction analysis are not feasible and quick and easy application tools are required. In this regard, our study contributes to the existing literature by providing data on a range of parental behaviors referring to the various dimensions that have been associated with optimization of language development using the PICCOLO, a reliable and valid tool. As we have mentioned, most analyses in the literature focus on a particular set of behaviors grouped in ad hoc categories. The Parenting Interactions with Children: Checklist of Observations Linked to Outcomes (PICCOLO) [50,51] has proven useful to assess 29 types of parental behavior that have been linked to child developmental outcomes (for a review, see Roggman et al. [50,51]), including affective (affection) and responsive behavior (responsiveness), non-intrusiveness (encouragement) and child’s language and reasoning stimulation (teaching). PICCOLO is fully recognized as a psychometrically strong tool for measuring developmentally supportive parent–child interactions. It meets the criteria of good validity, reliability and appropriateness for research and for use in early intervention practice [52,53]. 

The Spanish version of PICCOLO has shown to be useful to assess mothers’ and fathers’ parental behavior when they interact with children with disabilities [54] and with typically developing children [55]. Spanish mothers and fathers show very similar strengths to support child development when they interact with their children during play, when children are typically developing [55] and when they have intellectual disabilities [54]. Research using PICCOLO has shown that caregiver responsiveness predicts higher child communication levels [56]. In addition, in a previous study with a Spanish sample of mothers and fathers with children with intellectual disabilities, mothers’ responsive behavior and fathers’ teaching behavior predicted the children’s linguistic outcomes at early ages [57]. All this evidence reinforces the idea that the type of parenting exercised by the main caregivers may promote language development in typically developing children [25,27] and children with some type of developmental delay [7,58,59]. This is further reinforced by studies that have shown improvements in children’s language development because of parenting-focused interventions [60,61,62,63].

Since most studies on the relations between parental behaviors and child language have been carried out in the United States, it is interesting to provide data from a European country such as Spain. Participation of mothers and fathers in parenting tasks in present day Spain is far more balanced than in the past. Therefore, there are reasons to think that both mothers and fathers can make positive contributions to language development of their children. Even so, the model of the father as the provider of support to the mother, the main caregiver, still predominates [64].

The aim of this study was to explore the relation between parenting behavior assessed with PICCOLO and child language development in typically developing children at early ages. We controlled for some sociodemographic variables. Our study contributes to the existing literature by providing data on a broad set of parenting behaviors using a reliable and valid tool in a sample that includes both mothers and fathers from the same families in Spain.

## 2. Materials and Methods

### 2.1. Participants

Participants were 90 children, 88 mothers and 76 fathers (who answered PICCOLO), 74 of them from the same families. All of them were living in Spain. The children were aged between 15 months and 31 months (M = 24.57, SD = 3.99). Forty-one percent were younger than 2 years old (15 to 23 months); 59% were 2 years old (24 to 31 months). Forty-nine were male (54%) and 41 were female (46%). All children were born in Spain. Only two of them were preterm and required specialized medical attention at delivery. At the time of the study, all the children were healthy and typically developing, as determined by their primary care pediatric history. According to the Bayley Scales of Infant Development (BSID-III) administered in this study, the child’s percentile scores ranged from 25 to 100 for cognition (M = 119.67, SD = 17.20, range 90–145). The Bayley’s mean and SD reflect the age-standardized score (M = 100; SD = 15, range 50–150). All subjects scored above the basal limit [49], which indicates cognitive development in the normal range at the time of the study.

The mothers were aged between 29 and 46 years (M = 36.67, SD = 4.08) and 90% were from Spain. Seventy-seven percent of the mothers had a university degree and 23% did not. Most were in full-time (45%) or part-time employment (44%), whereas 11% were not in paid employment. In 88% of cases (n = 79), the children’s mothers were married or cohabiting, in 8% (n = 7) they were single, divorced, separated or widowed and in four cases no information was available. 

The fathers were aged between 26 and 56 years old (M = 38.95, SD = 5.76) and 93% were from Spain. Sixty-three percent of the fathers had a university degree and 37% did not. Most of them were in full-time employment (94%); the rest were employed part-time (4%) or unemployed (2%). For ninety percent of the children (n = 80), their fathers were married or cohabiting, one father (1%) was divorced and for 10% (n = 9) of the children we have no information about their parents’ civil status. 

Family languages were Catalan and Spanish (bilingual) (33%), Catalan (31%), Spanish (20%) and others (17%). Thirteen percent of the families had a monthly income between EUR 1602 and EUR 2451, which is considered an average income in Spain. Six percent of families had a monthly income below EUR 1602 and 73% had a monthly income above EUR 2451.

### 2.2. Instruments

Sociodemographic data were collected by applying a questionnaire asking for data referring to the child (age, gender, attendance of a nursery or preschool center) and the parents (gender, age, educational level, working status, civil status and monthly family income).

Child development was assessed with the Spanish version of the Bayley Scales of Infant Development-III (BSID-III) [65]. BSDI-III scales are widely used to assess development between 1 and 42 months of age. Children are assessed in the five key developmental domains of cognition, language, social-emotion, motor and adaptive behavior. Cognition, expressive language and receptive language scales were applied in this study. The cognitive scale assesses children’s sensorimotor development, exploration and manipulation of objects, object relations, concept formation and memory. The expressive language scale assesses preverbal communication (e.g., babbling, gesturing), vocabulary use (e.g., naming objects and pictures) and morpho-syntactic development (e.g., two-word utterances, plurals). The receptive language scale assesses comprehension of preverbal behaviors, vocabulary, morphological units (e.g., pronouns, prepositions) and morphological markers (e.g., plurals, possessives). The scales were administered in accordance with the manual’s instructions. Items were scored “1” for correct and “0” for incorrect. For each scale, the starting point was determined by the child’s age. If the child scored “0” on any of the first three items, the examiner went back to the previous age’s starting point until three consecutive “1”s were reached. The administration finished when five consecutive “0”s were attained. Bayley-III has demonstrated high reliability and validity in Spain [65,66].

Parental behavior in adult–child interactions was assessed with the Parenting Interactions with Children: Checklist of Observations Linked to Outcomes [50,51]. PICCOLO has been validated in Spain with mothers [67] and fathers [68]. The PICCOLO is a 29-item observational measure of parent–child interactions for parents with children aged between 10 and 47 months (to revision). The 29 items reflect parental behavior linked to a child’s developmental outcomes. They are scored according to frequency and consistency as 0 (absent, not observed), 1 (rare, minor or emerging) and 2 (clear, definitive, strong and frequent). The items are grouped into four domains: (a) affection (7 items), which involves the physical and verbal expression of affection, positive emotions, positive evaluation and positive regard (e.g., “Shows emotional warmth”, “Smiles at child”); (b) responsiveness (7 items), which includes reacting sensitively to a child’s cues and expressions of needs or interests and reacting positively to the child’s behavior (e.g., “Changes pace or activity to meet child’s interests or needs”, “Follows what child is trying to do”); (c) encouragement (7 items), which considers the parents’ support of their child’s efforts, exploration, independence, play, choices, creativity and initiative (e.g., “Encourages child to handle toys” or “Supports child in doing things on his/her own”) and (d) teaching (8 items), which includes cognitive stimulation, explanations, conversation, joint attention and shared play (e.g., “Explains reasons for something to child”, “Labels objects or actions for child”). See Roggman et al. [50] or Vilaseca et al. [67] for a detailed list and description of the 29 PICCOLO items. The instrument generates a score for each dimension between 0 to 14 (and 0 to 16 for the teaching dimension) and a total score between 0 and 58. The psychometric properties of PICCOLO have been found to be satisfactory for the original and the Spanish version. In this study, two trained observers coded 15.6% of mother–child interactions and 9.23% of father–child interactions. Inter-rater reliability scores were good to excellent (ICCs from 0.78 to 0.99). Regarding total scale consistency reliability, Cronbach’s α values were 0.87 for mothers (n = 88) and 0.86 for fathers (n = 76). With respect to the PICCOLO subscales, Cronbach’s α value ranged between 0.51 and 0.73 for mothers and 0.54 and 0.76 for fathers. 

### 2.3. Procedure

Ethical approval was obtained from the Bioethics Commission of the first author’s university, according to the guidelines provided by the Council for International Organizations of Medical Sciences (CIOMS), the World Health Organization (WHO) and the World Medical Association (WMA) Declaration of Helsinki—Ethical Principles for Medical Research Involving Human Subjects. 

Families were recruited from pediatric centers, nurseries and community family centers. They were informed that their participation would be voluntary and anonymous. They received a written document with information about the study, the sociodemographic questionnaire and a brief guide containing basic recommendations to video record adult–child interactions during play at home. The parents provided signed informed consent prior to participation. Mothers and fathers were asked to self-record, separately, a session lasting between 8 and 10 min playing with their child at home, in their usual way and with their own toys, following the instruction: “Interact and play with your child as you normally do”. Some games and materials were suggested in the brief guide, for example, books, toy animals, toy kitchens, dolls, building blocks, etc. The father’s and the mother’s self-recordings could be made on the same or different days, within a maximum period of one week. No differences were observed in the type of toys chosen by mothers and fathers, and most adults and children introduced different toys during the play sessions. Finally, the videos were collected and scored according to the PICCOLO criteria by a group of trained psychologists and specialists in child development. Only videotapes that followed the researcher’s instructions (99%) were scored.

A child’s cognitive development was assessed with the cognition scale of Bayley-III [65] to control that the cognitive development of children was within normal limits. The receptive language and the expressive language scales were applied to assess the children’s linguistic development. Percentiles were obtained for the cognition scale. For the language scales, scalar scores were used to assess receptive and expressive language and the composed score to obtain a total language measure. Bayley’s scalar scores range from 1 to 19, with 10 as the average score and 3 the standard deviation. Composed scores range from 46 to 154, with 100 as the average score and a standard deviation of 15.

For every family, a maximum period of one month was established to collect the parents’ and the child’s measures.

### 2.4. Data Analysis

Bivariate analyses were carried out to identify potentially useful predictors to be included in predictive models. A significance level of 5% was used to keep predictors in further modeling steps. To assess the relationship between explanatory variables (affection, responsiveness, encouragement, teaching and total parenting) and responses (receptive language, expressive language and total language), parametric (*t*-tests, F tests and Pearson’s correlations) and non-parametric (bootstrap BCA (Bias-Corrected and Accelerated)with 1000 replications for estimating confidence intervals [CIs] for correlations) tests were employed. Once the set of potential predictors had been determined, linear regression models were estimated for each response of interest. These first estimated the models’ complexity (i.e., the number of predictors included in the model), which was reduced whenever possible using an information index, AIC (the lower the index, the better the model’s fit to the data) and confirmed with an L1-regularization (LASSO) procedure, using leave-one-out cross-validation (LOO-CV regularized models). The coefficients obtained in LOO-CV regularized models were employed to confirm that the final models included only predictors with relatively stable coefficients under the set of studied conditions. This is especially useful for models that might generalize well. All the assumptions of the linear models were checked (linearity, homoscedasticity and normality) by analyzing the models’ residuals. Concerning missing data handling, pairwise complete or complete observations procedures were used when bivariate associations were assessed and multiple regression models were estimated, respectively. All analyses were carried out using statistical software R version 4.2.1 (R Foundation for Statistical Computing, Vienna, and Austria) [69]. 

## 3. Results

Table 1 shows a numerical description of children’s language development scores (receptive language, expressive language and total language) and mothers’ and fathers’ scores in the parenting domains (affection, responsiveness, encouragement, teaching and total). 

Regarding parenting, the mean scores of mothers and fathers in all the PICCOLO domains and the total score corresponded to a middle range according to the Spanish normative values for mothers and fathers [70]. The mean scores for the four parenting domains followed a similar pattern in both mothers and fathers. The highest mean score was in the responsiveness domain, followed by the affection, encouragement and teaching domains.

Bivariate analyses showed that none of the sociodemographic parental factors included in the study (parents’ age, education, employment status and family’s income) were related to language development scores. All the observed effect sizes were small (see Table 2).

### 3.1. Parenting Behavior and Child’s Language Development

Table 3 summarizes the bivariate tests between PICCOLO subscales and language development scores. Some of the mothers’ parenting scores were found to be significantly associated with receptive language development: responsiveness (r = 0.22; *p* = 0.04; bootstrap BCA 95% CI = [0.05; 0.39]), encouragement (r = 0.26; *p* = 0.02; bootstrap BCA 95% CI = [0.03; 0.45]) and teaching scores (r = 0.30; *p* = 0.01; bootstrap BCA 95% CI = [0.06; 0.49]). As for the fathers’ parenting, encouragement (r = 0.35; *p* < 0.01; bootstrap BCA 95% CI = [0.15; 0.51]) and teaching (r = 0.25; *p* = 0.04; bootstrap BCA 95% CI = [0.04; 0.46]) scores were found to be related positively and significantly to receptive language. Concerning expressive language development, mothers’ responsiveness (r = 0.28; *p* = 0.01; bootstrap BCA 95% CI = [0.08; 0.45]) and teaching (r = 0.35; *p* < 0.01; bootstrap BCA 95% CI = [0.12; 0.54]) scores were positively related below the chosen significance level. The fathers’ teaching score was the only variable associated with expressive language development (r = 0.38; *p* < 0.001; bootstrap BCA 95% CI = [0.16; 0.54]). Finally, when bivariate analyses were performed for the total language development scores, responsiveness (r = 0.28; *p* < 0.01; bootstrap BCA 95% CI = [0.10; 0.44]), encouragement (r = 0.23; *p* = 0.04; bootstrap BCA 95% CI = [−0.01; 0.40]) and teaching (r = 0.38; *p* < 0.01; bootstrap BCA 95% CI = [0.15; 0.57]) mothers’ scores were found to be significantly related to language. In contrast, only fathers’ teaching scores were found to be significantly related to total language scores (r = 0.38; *p* < 0.01; bootstrap BCA 95% CI = [0.18; 0.54]).

### 3.2. Regression Models for Child’s Language Development

Separate multiple regression models for the three measures concerning language development were estimated by initially employing the variables with which the development scores were significantly associated. The initial models’ complexity was reduced by using an information criterion (AIC) to deal with a more parsimonious model that still fitted adequately the data at hand. Given the sample size and to provide models that might generalize well, we also checked that the final models included only predictors with stable coefficients. To that aim, LOO-CV-regularized models were employed to check that the coefficients for the predictors that were kept in the final models were stable (with values around those found in the previous linear models) and consistent (with no sign changes detected throughout the replications). Table 4 shows the summary results of the final linear models.

The linear model with children’s receptive language development scores as the dependent variable included mothers’ encouragement scores (β = 0.30, *p* = 0.01; 95% CI = [0.08; 0.53]) and fathers’ encouragement scores (β= 0.32, *p* = 0.01; 95% CI = [0.09; 0.54]) kept as predictors. This model had a related predictive capacity of about 18%.

The final model for predicting expressive language development scores included mothers’ responsiveness (β = 0.19, *p* = 0.11; 95% CI = [−0.04; 0.42]) and fathers’ teaching scores (β = 0.33, *p* = 0.01; 95% CI = [0.09; 0.56]) as predictors. The model that was finally kept explained about 16% of the variability in expressive language development.

Finally, the linear model for predicting total language development scores kept as predictors mothers’ responsiveness (β = 0.20, *p* = 0.09; 95% CI = [−0.03;0.43]) and fathers’ teaching scores (β = 0.32, *p* = 0.01; 95% CI = [0.08;0.55]). This model explained about 16% of the total variability in total language development.

## 4. Discussion

The overall aim of this study was to examine the relationship between parenting behavior and child language development in typically developing children at early ages, with some sociodemographic variables controlled.

Contrary to what is established in the literature, in our sample, parents’ educational level and family income did not show any relation to children’s language development assessed with Bayley-III at early ages. This could be, in part, a result of the relative homogeneity of our sample with respect to these variables of SES. Most of the mothers and fathers were university graduates and had a salary above average in Spain. Accordingly, our results with respect to the relations between parenting and child’s language development should be considered to refer to the Spanish population of medium–high educational and economic levels. 

### 4.1. Parenting Behavior and Child Language Development

With respect to positive parental behavior linked to child development, previous studies with PICCOLO in Spain have not shown differences according to a child’s gender [55], as was the case in this study. With respect to parents, mothers scored higher than fathers in all parenting domains but showed very similar patterns and strengths. Mothers and fathers were competent in all parental domains, scoring in the medium ranges and higher in responsiveness, followed by affection, encouragement and teaching in the aforementioned and present studies. Accordingly, responsiveness would be the strongest point of Spanish mothers and fathers and teaching the lowest. Notably, teaching is the dimension that showed the highest standard deviations, which indicates more interindividual variability. Teaching was also the weaker parental domain in previous studies that used PICCOLO to assess mothers’ and fathers’ parenting with his/her son/daughter with intellectual disabilities [54], which is a relevant fact to inform family programs. 

In accordance with what is established in the literature about the relations between parents’ behavior when they interact with their children and a child’s linguistic outcomes at early ages, our results showed significant positive correlations, at low and moderate levels, between three of the four parenting dimensions of the PICCOLO (responsiveness, encouragement and teaching) and the child’s language development. 

With respect to responsiveness, our results showed significant positive correlations, at low and moderate levels, between mothers’ responsive behavior and a child’s expressive, receptive and total language scores. A large body of literature has shown positive relations between maternal responsive behavior and typically developing children’s outcomes at early ages [22,23,24,25,26]. Previous studies that assessed responsiveness with PICCOLO have shown a positive relation between a teacher’s responsive behavior [56] and a child’s communicative acts in early care and education settings and positive relations between maternal responsiveness and cognitive and linguistic development in children with intellectual disabilities [57,71]. Our results coincide with previous studies that show positive relations between maternal responsiveness and a child’s language development. The types of responsive behavior assessed with PICCOLO include “Pays attention to what child is doing”, “Follows what child is trying to do” or “Replies to child’s words or sounds”, which indicate contingent responses by the adult to the child interests, motivations and behavior. As mentioned in the Introduction, a large body of literature has shown that contingent responses to a child’s focus of attention and action benefit the child’s linguistic outcomes [22,28,29]. As Tamis-LeMonda [27] has pointed out, contingent responses help the child to establish relationships between language and the world and provide linguistic models. Furthermore, frequent inclusion of non-verbal support, such as gazes and gestures, facilitates language learning. Additionally, we can point out that responsive behavior clearly benefits establishment of joint attention, which in turn is especially relevant to support communicative and linguistic development [72,73]. 

Regarding fathers’ responsiveness, no relations with child’s language have been found. Previous studies using PICCOLO did not find a relation between fathers’ responsiveness and language development in children with intellectual disabilities [57,71]. Some researchers have reported lower levels of fathers’ responsiveness than mothers [74,75]. Other researchers have not identified these differences [76,77]. In our study, fathers scored very similarly to mothers in responsiveness. However, a recent review [78] concluded that there is more evidence in favor of fathers showing less responsive behavior than mothers when they interact with their children at early ages. Furthermore, although some studies have reported positive relations between fathers’ responsivity and child linguistic outcomes [77], most of the studies about these relations have been conducted with mothers. Although some studies have reported that both mothers’ and fathers’ responsiveness positively relate to a child’s cognitive and linguistic outcomes [79], more research is needed on this topic.

In our study, affective behavior by itself was not related to a child’s language development. In a review of the literature about parental behavior and child development, we did not find studies that directly related parental affective behavior and child’s language development. Affection has been more closely related to a child’s social and adaptative behaviors and to cognitive development (for a review, see Roggman et al. [50,51]. However, as many authors have pointed out [22,32], a parent’s responsiveness or sensitivity commonly occur in a positive affective and emotional climate. Perhaps a parent’s affective behavior by itself and isolated is not the most important factor to support a child’s communicative and linguistic outcomes. It may be relevant to the child’s linguistic trajectory when it is linked to other behavior that is more specifically related to language development, such as responsive behavior. 

As expected, encouragement, which includes non-directive behavior, such as “Waits for child’s response after making a suggestion”, “Supports child in making choices” or “Verbally encourages child’s efforts”, has been found to be positively linked to a child’s language. In this case, a positive correlation was found for mothers and fathers between encouragement and a child’s receptive language at low and moderate levels, respectively. Mothers’ encouragement was also related to a child’s total language score, with a low-level positive correlation. Our results, which indicate a positive relation between non-directive and non-intrusive parental behavior and a child’s linguistic outcomes, are in line with those of many previous studies about directiveness and child language [37,38,39,40]. When mothers and fathers are encouraging, children are more prone to take on challenging tasks and develop better cognitive, linguistic and social skills [80].

Teaching was the parenting dimension that showed more significant positive correlations with a child’s linguistic outcomes. For mothers, teaching correlates at moderate levels with a child’s receptive, expressive and total language scores. For fathers, the same correlations were identified, at a low level for receptive language and at a moderate level for expressive and total language. The teaching PICCOLO subscale includes behavior such as “Explains reasons for something to child”, “Repeats or expands child’s words or sounds”, “Labels objects for the child”, “Talks about characteristics of objects”, “Engages in pretend play with child” or “Asks child for information”. These expected results are very consistent with previous literature as it is well-established that the way adults structure their conversational interchanges with children and some characteristics of parental child- or infant-directed speech are relevant to a child’s linguistic outcomes [10,43,44]. As mentioned previously, richness of vocabulary and parental behavior as logical explanations and symbolic play have been shown to be related to language development [43]. This is also the case for questions [48,49], repetitions of the child’s vocalizations or utterances [45,46] and syntactic or semantic expanding of the child’s utterances [47]. In other studies with families with children with intellectual disabilities, positive relations between teaching and a child’s linguistic outcomes have been found, but only for fathers [57,71].

### 4.2. Regression Models on Bayley Language

Mothers’ and fathers’ encouragement scores predicted 18% of the children’s variability in receptive language, as assessed by BSID-III. Additionally, mothers’ scores on responsiveness and fathers’ scores on teaching predicted 16% of the variability in both a child’s expressive language and the total language scores. Accordingly, parental behavior assessed with PICCOLO has some predictive value for child’s language in typically developing children at early ages. This study provides some new data, from the Spanish context, about the relevance of positive parenting behavior and a child’s linguistic development.

One relevant issue is that the relations were given specifically with the child’s receptive language in the case of mothers’ and fathers’ encouragement scores and with expressive language for mothers’ responsiveness and fathers’ teaching scores. 

In relation to receptive language, we must consider that PICCOLO’s encouragement behavior refers to active support of a child’s initiative, creativity and play. When parents are encouraging, children may show more persistence on tasks that are difficult or challenging, and this may be related to better cognitive and language outcomes. In addition, parents’ verbal encouragement could be related to better executive function [81], and this may be related to better levels of receptive and expressive language [82].

The associations between expressive language and mothers’ responsiveness and fathers’ teaching scores are interesting, especially if we consider co-parenting for promotion of child development. Teaching refers to cognitive and linguistic stimulation and is especially relevant to optimal linguistic development, as is responsiveness, as previously explained in the Introduction. Both responsiveness and teaching behavior promote language development outcomes [83].

As mentioned above, previous studies assessing parental interactions with children with intellectual disabilities, using PICCOLO, found the same predictive relations between mothers’ responsiveness, fathers’ teaching and the child’s language development assessed with Bayley-III [57,71]. These predictive relations were also found for child’s scores in cognition. In these studies, with children with intellectual disabilities, parents’ encouragement did not correlate with a child’s language or cognition measures. Unlike the present study, only the total scores for language development were considered. It would be interesting to analyze the relations between PICCOLO’s domains and the specific measures of receptive and expressive language to assess whether the relationships identified in this study are similar for children with intellectual disabilities. 

There may be specific relations between some domains of parental behavior and components of a child’s language development for which we do not currently have a plausible explanation. Beyond this, the fundamental contribution of our study is to provide new data about the relationships between positive parental behavior in interactions with their children at early ages and a child’s linguistic outcomes. Several studies have shown that good parent–child interactions also benefit language development in children with disabilities [7,58,71]. Creating a climate of affection and emotional warmth when interacting with the child, being attentive to his/her signals (emotional states, motivations and interests), interpreting them properly and giving a contingent response, promoting the child’s initiative, effort and autonomy as well as encouraging conversation and reasoning are key features of positive parenting—the kind of parenting that contributes to child development in both typical and atypical populations.

Our data are of interest to inform evidence-based professional practice from naturalistic models for families of children who have language development delays or disorders. While many parents naturally exhibit behaviors that define quality parenting interactions, many others require professional intervention to optimize their daily interactions with their children, especially when they are in a situation of social vulnerability and/or the children have developmental delays or disorders. In this context, interventions would be directed at collaboratively working with mothers and fathers to increase in play and other daily routines regarding PICCOLO’s behaviors that are positively related to a child’s language development. We recommend that these interventions be framed in a family-centered approach, working together with families to find strategies to improve the quality of parental interactions in daily life. Interestingly, when professionals facilitate natural learning opportunities at home, parents are more likely to report positive outcomes, including better family well-being and greater child development progress. Video-feedback interventions for parents, which increased during the pandemic [84], represent a very promising initiative that can provide parents with opportunities to promote positive parenting behaviors, with potential benefits for their own emotional well-being and for their children’s development [85]. 

### 4.3. Limitations and Final Remarks

The present study extends the current literature on parenting and child language development in families of young children with typical development. Nevertheless, it has several limitations that should be considered when the results are interpreted. First, the sample was not probabilistic. Parents who agreed to participate in this study were probably the most informed and involved in childrearing and education. Additionally, most of the mothers and fathers had a university degree and a family income above the average in Spain. These factors, together with the appropriate developmental levels of the children, are probably the reason for the relatively high average scores on the PICCOLO [14,15,16]. Another limitation is that a cross-sectional correlational design was employed, so we cannot establish clear causality between predictors and responses. In this regard, we need to be cautious about use of the term “predictor” in the regression analysis since it does not necessarily imply direct causality but covariation. Given the effective sample size in this study, a likely risk is overfitting, that is, including too much complexity in the model. This may mean that the model does not generalize well (for instance, with a different dataset). To avoid this problem, an LOO-CV procedure was carried out, which allowed us to provide stable estimates for the coefficients (i.e., those consistently different from zero).

An aspect to consider that could be of interest in future studies refers to family variables that could explain parents’ behavior when they interact with a child. In this study, we focused on a limited set of parents’ variables (parents’ age, education, employment status and family income). However, other factors, such as parents’ beliefs, knowledge and attitudes toward development and education, could be relevant to explain parental behavior and its impact on child language development. Furthermore, more variability with respect to SES variables should be included.

Future research about mothers’ and fathers’ parenting behaviors and their relationship with a child’s language development should include recorded observations of mother/father–child interactions at later ages and an analysis of the possible continuity of mother/father–child interaction patterns and their links to children’s developmental outcomes in early school years.

## Figures and Tables

**Table 1 children-10-00505-t001:** Description of numerical factors related to parenting domains and children’s development.

Participant	Factors	Range	M	SD
Children (n = 90)	Total language score	79–153	107.40	16.13
	Expressive language score	3–19	10.37	3.32
	Receptive language score	6–19	12.14	3.01
Mothers (n = 88)	Affection	6–14	11.82	1.78
	Responsiveness	7–14	12.52	1.83
	Encouragement	5–14	11.25	2.48
	Teaching	4–16	11.24	2.73
	Total parenting	24–58	46.83	6.87
Fathers (n = 76)	Affection	6–14	11.43	1.90
	Responsiveness	6–14	12.04	2.02
	Encouragement	2–14	10.50	2.85
	Teaching	2–16	10.57	3.07
	Total parenting	20–58	44.53	7.31

**Table 2 children-10-00505-t002:** Summary table with association tests between sociodemographic factors related to parents’ characteristics and child’s language development scores.

	BSID-III Scores		
	Receptive Language	Expressive Language	Total Language
Mother’s			
Age	r = 0.14; *p* = 0.21	r = 0.07; *p* = 0.53	r = 0.08; *p* = 0.5
Education	t(79) = −0.18; *p* = 0.86; d = 0.05	t(78) = 0.01; *p* = 0.99; d = 0.0	t(80) = −0.44; *p* = 0.66; d = 0.12
Employment	F(2,78) = 0.07; *p* = 0.93; η^2^ = 0.00	F(2,77) = 1.0; *p* = 0.37; η^2^ = 0.03	F(2,79) = 0.56; *p* = 0.58; η^2^ = 0.01
Father’s			
Age	r = -0.01; *p* = 0.91	r = 0.0; *p* = 0.98	r = −0.03; *p* = 0.79
Education	t(71) = −1.26; *p* = 0.21; d = 0.31	t(71) = 0.24; *p* = 0.81; d = 0.06	t(73) = −0.41; *p* = 0.68; d = 0.1
Employment	F(2,73) = 0.33; *p* = 0.72; η^2^ = 0.01	F(2,72) = 1.87; *p* = 0.16; η^2^ = 0.05	F(2,74) = 1.34; *p* = 0.27; η^2^ = 0.03
Family			
Monthly income	F(2,76) = 1.48; *p* = 0.24; η^2^ = 0.04	F(2,75) = 0.71; *p* = 0.49; η^2^ = 0.02	F(2,77) = 1.21; *p* = 0.3; η^2^ = 0.03

Note: r = Pearson’s correlation tests; t = two-sample *t*-tests and F = one-way ANOVAs.

**Table 3 children-10-00505-t003:** Bivariate correlations between parenting scores and child’s language development scores.

	BSID-III Scores		
	Receptive Language	Expressive Language	Total Language
Mother’s			
Affection	0.11	0.01	0.04
Responsiveness	0.22 *	0.28 *	0.28 **
Encouragement	0.26 *	0.19	0.23 *
Teaching	0.30 **	0.35 **	0.38 ***
Father’s			
Affection	0.10	0.12	0.11
Responsiveness	0.16	0.13	0.17
Encouragement	0.35 **	0.11	0.23
Teaching	0.25 *	0.38 ***	0.38 ***

Note: * < 0.05, ** < 0.01,*** < 0.001.

**Table 4 children-10-00505-t004:** Summary of multiple linear regression models for predicting child’s language development.

	Estimate	SE	Standardized Beta	t	*p*-Value
Receptive Language					
Intercept	4.23	2.08		2.04	0.05
Mother’s Encouragement score	0.38	0.14	0.30	2.67	0.01
Father’s Encouragement score	0.36	0.13	0.32	2.78	0.01
Adj. R2 = 0.18					
Expressive Language					
Intercept	1.75	2.85		0.62	0.54
Mother’s Responsiveness score	0.38	0.23	0.19	1.62	0.11
Father’s Teaching score	0.37	0.13	0.33	2.80	0.01
Adj. R2 = 0.16					
Total language					
Intercept	65.58	13.58		4.83	< 0.001
Mother’s Responsiveness score	1.90	1.10	0.2	1.72	0.09
Father’s Teaching score	1.70	0.63	0.32	2.72	0.01
Adj. R2 = 0.16					

## Data Availability

Not applicable.

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
