# Peer review of "Relations between Positive Parenting Behavior during Play and Child Language Development at Early Ages"

_children, 2023, doi:10.3390/children10030505_

Round 1

Reviewer 1 Report

A limitation section should be added before the publication.

Overall a well written article.

Author Response

We thank the reviewer for the positive evaluation he has made of our manuscript.
We have incorporated the reviewer's suggestion. Limitations were included in the “Final remarks” section. For better identification, we have changed the section title to “Limitations and final remarks”

Reviewer 2 Report

This paper assesses positive parenting behaviors using the Parenting Interactions with Children: Checklist of Observations Linked to Outcomes and child language development among typically developing children in Spain.

I have two major concerns. The first is that there is no written justification for the study. As written, there is not a clear gap that this research is filling. In fact, the literature review makes it sound like it is similar to some existing studies, without specifying the differences. The second concern is related, because there is no established gap, I am unsure that this manuscript is contributing to the existing literature.

I would like to know more about the context of parenting behaviors and language development in Spain. This is crucial information in order to understand the findings.

The literature review seems to cover the most important literature. However, there are some excessively wordy sentences such as page 3 lines 19-21 and overly complicated language.

Readers may not be as familiar with the Bayley Scale, so a bit more information about what it is measuring would be helpful. Additionally, an appendix table with the PICCOLO and Bayley measures/questions would be a useful addition. These additions would help to clarify what type of language acquisition is being measured. These are young kids, so specifying what skills are of interest would be helpful.

I am not familiar with the PICCOLO measurement, as such, I am concerned about how representative the recorded interaction is with regular, unrecorded interaction. I imagine parents would be on their best behavior during the recording. Addressing this concern is important to validate the findings.

More ‘so what’ recommendations for families and more about the role of similar educational levels and SES. As mentioned in the limitations, this group of high SES, highly educated parents is particularly invested in parenting practices that are captured with this measurement. What does that mean for the findings and the policy or practice that you recommend.

Author Response

We would like to thank the reviewer for their comments. We accept all his/her suggestions and have revised the manuscript accordingly. A table is shown in the attached document. In the first column of the table, we list the reviewer's comments, and in the second column we respond to the comments.

Round 2

Reviewer 2 Report

Thank you for these revisions. They help explain the gap in the literature, the role of Spain in this research, and provide a 'so what' in multiple domains. 

There are occasional typos in the highlighted (new) sections that need to be addressed, in addition to the manuscript at large. Re-reading for minor editing would be beneficial. 

Please include a citation for lines 170-171 about parenting behaviors in Spain. Even adding one more sentence explaining how parenting behavior in Spain may differ from parenting in the United States, where most of the research has been carried out, would be helpful here. 

Thank you for adding more information about the scales, it is very useful information. 

Author Response

We would like to thank the reviewer for their comments. We accept his/her suggestions:

-We have re-read the text and we have corrected some typographical errors.

-We have added a little more detail about the Spanish context for parenting in Spain, and a citation about parental roles in this country (lines 173-174).

See attached file. Thank you.
